# An 11-yr (2007 – 2017) soil moisture and precipitation dataset from the Kenaston Network in the Brightwater Creek basin, Saskatchewan, Canada.

Erica Tetlock[1], Brenda Toth[1] Aaron Berg[2], Tracy Rowlandson[2], Jaison Thomas Ambadan[2]

[1] Environment and Climate Change Canada, Saskatoon, Saskatchewan, Canada
[2] Department of Geography, Environment and Geomatics, University of Guelph, Guelph, Ontario, Canada

*Correspondence to*: Erica Tetlock (erica.tetlock@canada.ca)

**Abstract.** Soil moisture and precipitation have been monitored in a hydrometeorological network situated within the Brightwater Creek basin, east of Kenaston, Saskatchewan, Canada, since 2007. The majority of the prairie landscape is annually cropped with some sections in pasture. This agricultural region is ideal for remote sensing validation and calibration and, in conjunction with the flux tower situated within the network, hydrological model validation. Remote sensing validation collaborations have included European Space Agency's Soil Moisture Ocean Salinity (SMOS) and NASA's Soil Moisture Active Passive (SMAP). The network was developed at two spatial scales, one high-resolution set of sites installed over a 10 km × 10 km region and a second installed over 40 km × 40 km. The sites are all similar in design with three instrument depths for soil moisture and temperature, as well as precipitation measurement. The 2007 – 2017 dataset published in this paper has gone through a quality control review process, which involved both automated and manual processes. The dataset is limited to the summer months (May 1 – Sept 30) due to the uncertainties and complexities of measurement in frozen soils and the freeze/thaw period each year. Data discussed in this publication are available at https://dx.doi.org/10.20383/101.0116, and data beyond 2017 can be requested from the corresponding author.

## 1    Introduction

Soil moisture and precipitation are important elements of the hydrological cycle. While soil moisture constitutes a small portion of the global water cycle, it has a significant influence on atmospheric and hydrologic processes. Soil moisture is highly variable across a landscape, being influenced by both atmospheric conditions (e.g. precipitation, evaporation), landscape variability (e.g. topography, soil characteristics), and vegetation. This creates difficulty when attempting to asses soil moisture at the typical scales of atmospheric circulation models (Crow et al., 2012), however inclusion of soil moisture as a dynamic parameter within numerical modelling

improves forecast skill for both hydrological and meteorological models (Koster et al., 2010; Koster et al., 2011; Drewitt et al., 2012; Wanders et al., 2014). The difficulty of measurement has prompted researchers to develop

remote sensing techniques to try and quantify soil moisture conditions at various scales. Any remote sensing technique requires calibration and validation, in this case achieved with *in situ* monitoring stations.

Relatively few monitoring network exist across the Canadian Prairies and the variation in landscape and climate present particular challenges. Other networks include the Agriculture and Agri-Food Canada (AAFC) network in Manitoba (Bhuiyan et al., 2018) and the stations established across the agricultural regions of Alberta (Walker

and Howard, 2003), along with the Kenaston Network in Saskatchewan. The Kenaston Network was designed to fulfil both the needs of land-atmospheric modelling and remote sensing validation programs. Specifically for remote sensing of soil moisture, the individual stations were distributed at two spatial scales to accommodate validation of remote sensing products at various scales. The high resolution of the network sites allows for validation of remote sensing products or hydrological models at a range of spatial scales.

To date, the network has been widely used for several purposes in remote sensing hydrology (e.g. Chan et al., 2016), data assimilation (Dumedah et al., 2011; Reichle et al., 2017) and to a lesser amount in hydrological modelling (Garnaud et al., 2016). With respect to soil moisture remote sensing, validation studies have been performed for soil moisture retrievals derived from the Advanced Microwave Scanning Radiometer –Earth Observing System (AMSR-E) (Champagne et al., 2010) and retrievals derived from the AMSR-2 (Bindlish et al.,

2018). Further it has been used for validation of soil moisture retrievals from the Soil Moisture and Ocean Salinity mission (e.g. Champagne et al., 2016; Djamai et al., 2015) and the Soil Moisture Active Passive mission (e.g. Chan et al., 2016; Colliander et al., 2017) largely demonstrating statistically significant correlations to observed soil moisture anomalies. To continue the development of new applications and opportunities that make use of soil moisture data for this environment, the release and description of the collected soil moisture and precipitation

data sets to the broader public is of importance, and the purpose of this paper.

## 2   Network Description

The Kenaston Network, also called the Brightwater Creek Monitoring Network is located on the Canadian Prairies in central Saskatchewan, approximately 80 km south of Saskatoon. Stations within the network were established

in 2007 and consist of a series of soil moisture and precipitation sites, set at two spatial scales, and a year-round eddy-covariance tower with a full complement of meteorological instrumentation. The monitoring sites are situated within the basin of Brightwater Creek, which drains northward into the South Saskatchewan River.

Brightwater Creek has been monitored by a Water Survey of Canada flow gauge since 1965. The landscape is a typical prairie agricultural region with annually cropped fields, mainly of cereals, oilseeds, and pulse crops, and pasture lands. There are no irrigated sections in the study area, the nearest being the South Saskatchewan River District to the west surrounding Outlook, Sk. The area is flat with slopes of less than 2% (Burns et al., 2016) which affects runoff in the region. Significant portions of the area are considered non-contributing, where typically water does not drain to streams or rivers but instead ponds in small wetlands and sloughs (Shook et al., 2013). Texture of the soils in the region is predominantly silt loam but ranges from sandy loam to clay (Ellis et al., 1970, Magagi et al., 2013).

Data from the network have been used for several projects including the European Space Agency's (ESA) Soil Moisture and Ocean Salinity (SMOS) mission, the National Aeronautics and Space Administration's (NASA) Soil Moisture Active Passive (SMAP) mission, the Drought Research Initiative (DRI), and the Changing Cold Regions Network (CCRN). A field campaign for the SMAP satellite was conducted in 2010 (CanEx-SM10), primarily described in Magagi et al. (2013). Additional publications that describe the spatial scaling of the network include Rowlandson et al. (2015), and Burns et al. (2016).

The Kenaston Network is a community site, with involvement from Environment and Climate Change Canada (ECCC), the University of Guelph, the University of Saskatchewan, and AAFC, each of which is responsible for portions of the overall network. There are four AAFC stations, which are located within pasture sections and measure soil moisture down to 150 cm, along with standard meteorological sensors: data and site details can be found at [http://agriculture.canada.ca/SoilMonitoringStations/index-en.html]. This paper presents data only from the soil moisture and precipitation stations managed by Environment and Climate Change Canada and the University of Guelph and does not include data from the AAFC sites or the eddy-covariance tower managed by ECCC and the University of Saskatchewan. As mentioned above AAFC data is available through their website, and the eddy covariance tower data is in progress to be published. As of this publication a majority of the stations within the network are still operational and additional data can be requested from the corresponding author.

### 3    Soil Moisture and Precipitation Site Details

The soil moisture and precipitation sites are distributed at two spatial scales: 10 km × 10 km and 40 km × 40 km (Figure 1). The larger scale network has been modified over time and began in a 45 km × 55 km area, and correspondingly the number of sites has changed. Each site consists of a datalogger, power system, tipping bucket rain gauge (TBRG), and 3-4 Hydra Probes. These sites are usually set outside of the actively managed area of the cropped field, in fence line strips, under powerlines, or at the very edge of the field. There are two types of sites,

3-probe at the 40 km $\times$ 40 km scale, and 4-probe at the 10 km $\times$ 10 km scale. Figure 2 shows a typical setup for either type, with Figure 3 and Figure 4 clarifying the differences between the 3-probe and 4-probe sites, respectively. All sites have at least three probes, inserted horizontally at depths of 5, 20, and 50 cm below the surface that remain in place throughout the year. The 3-probe sites have all probes located at the edge of the field, outside of the actively managed field area. The 4-probe sites have a 5 cm probe at the edge of the field, with the 20 and 50 cm probes installed in the field, and a vertically placed probe, generally indicated as 0-5 cm, which is moved into and out of the field during the cropping season. The vertical probe is moved into the field after seeding and removed shortly before harvest and reinserted at the edge of the field for the off season. This movement of the vertical probe creates separate data streams, which have been separated in the data files to avoid confusion.

Data is collected at 30 minute intervals, a single point measurement from each Hydra Probe and the sum over the 30 minute interval for the TBRG. Provided from each probe for this dataset are real dielectric constant (real dielectric permittivity, $\varepsilon_r$), temperature, and soil moisture using the manufacturer's loam calibration equation. Additional data has been collected at some sites within the Kenaston network, including soil conductivity, 2.5 cm soil temperature, crop types, heights, and photos, air temperature and relative humidity, point measurement snow depth, and snow surveys, which is not included in this dataset but can be requested through the corresponding author.

Sites are visited regularly throughout the field season to ensure TBRG cleanliness and to check for site issues. Depending on the site these visits can be every two weeks or at minimum one a month, during the summer months. Sites with a vertically placed probe are visited more frequently than others due to the greater risk for disturbance and placement issues, with visits generally completed every two weeks.

### 3.1 Soil Instrumentation

The instrument used throughout the network to measure soil parameters is the Stevens Hydra Probe II (Stevens Water Monitoring Systems Inc, 2018a). These are radiometric coaxial impedance dielectric reflectometer sensors, with four 5.7 cm tines extending from a 3.4 cm diameter head, along which a radio frequency is applied and the reflected frequency measured (Stevens Water Monitoring Systems, Inc., 2018b). This reflected signal is related to the real dielectric constant ($\varepsilon_r$) of the soil which in turn is correlated to soil water content (e.g. Topp et al., 1980; Campbell, 1990; Seyfried et al., 2005). General ranges for $\varepsilon_r$ are roughly 80 in water, 1 in air, and 2-5 in dry soil. A more detailed description of the instrument and the measurement principles can be found in publications from Stevens Water Monitoring Systems, Inc. (2018a, 2018b). These sensors are widely used in university and government research networks, including NOAA's Climate Reference Network (Bell et al., 2013), the USDA's

Soil and Climate Analysis Network (Schaefer et al., 2007), and AAFC's national monitoring networks (Adams et al., 2014).

Real dielectric constant ($\varepsilon_r$) is related to soil moisture through a calibration equation (1) (Seyfried et al., 2005). The standard loam equation supplied by the manufacturer, with coefficients A = 0.109 and B = -0.179, report a sensor accuracy of ±0.03 $m^3m^{-3}$ (Stevens Water Monitoring Systems, Inc., 2018a or b), however a site specific calibration is recommended (e.g. Huang et al., 2004; Seyfried and Murdock, 2004; Rowlandson et al., 2013). The uncertainty in calibration method and ongoing work in this area presents a difficulty that has not been satisfactorily

resolved, particularly for the measurements at deeper depths, as described in Burns et al. (2014). To ensure consistency for all of the data the manufacturer supplied loam calibration equation (Stevens Water Monitoring Systems, Inc., 2018b) is used to calculate soil moisture, with the understanding that this decreases the overall accuracy of the network. Burns et al. (2014) reported loam calibration root mean squared errors (RMSE) ranging from 0.038 to 0.144 $m^3m^{-3}$, with improvements in RMSE when developing site specific calibrations. There have

been difficulties, however, in the repeatability of these site specific calibration methods and further work is required before applying site specific equations wholesale (e.g. Rowlandson et al., 2018). *In situ* calibration equations have been established for the majority of the near surface probes (5 cm) and while not used on the data for this paper these equations are available upon request.

$$\theta = A\sqrt{\varepsilon_r} + B \tag{1}$$

Occasional measurement issues with the Hydra Probe were encountered, some of which may be specific to the Kenaston network. For example, during hot summer days when the surface soil becomes very dry, $\varepsilon_r$ from the near surface probes (vertically placed 0-5 cm and horizontally placed 5 cm) will drop below ~2.6968, which produces a negative soil moisture value using the loam equation. These low $\varepsilon_r$ values are possibly due to soil cracking, poor sensor contact with the soil, or are simply valid responses from the probe. During these dry periods

repositioning the probe, which is the typical response to these types of issues in near-surface probes, is not typically possible simply due to the difficulty in inserting a probe into dry, hard-packed, fine grained soils. New cracks often form as the probe is taken out and re-inserted, resulting in the same issues. These probes are closely monitored and after the next sufficiently significant rain event, soil moisture typically increases and the probe begins responding as expected. Negative soil moisture values are automatically removed from the data set and

periods of prolonged data intermittence are also manually removed. Additionally, a diurnal oscillation of measured $\varepsilon_r$ is observed, with greater amplitude during hot, dry conditions. This suggests a temperature effect on $\varepsilon_r$ but is not investigated further here (Seyfried and Grant, 2007). Periods with significant diurnal oscillation and unrealistic soil moisture values are removed from the dataset.

The Kenaston region is similar to other parts of Saskatchewan in the occurrence of saline soils, the results of which cause some issues with the deeper probes (horizontally placed probes at 20 and 50 cm) (Seyfried and Murdock, 2004). While a typical variation between successive measurement intervals (timestamps) outside periods of rainfall could be on the order of $\pm0.01$ $m^3m^{-3}$, those probes measuring in saline conditions can vary as much as at $\pm0.10$-$0.20$ $m^3m^{-3}$. This is corroborated by measurement of soil conductivity: increasing variability between consecutive timestamps coincides with an increase in conductivity, generally greater than 0.2 S $m^{-1}$, which is less than the threshold given by the manufacturer of 1 S $m^{-1}$ (Stevens Water Monitoring Systems, Inc., 2018a). In some cases this only occurs for a season, while other sites show a consistent record of high conductivity and therefore large measurement variation in soil moisture. This type of issue can in certain cases be resolved by averaging the 30-minute data over a longer period, which is a common step used by modelling and remote sensing validation projects. Due to this, some periods of significant variation have been removed from the data set, however not all have been removed and should be reviewed by data users.

### 3.2    Precipitation Instrumentation

All sites within the network are equipped with a tipping bucket rain gauge (TBRG) to capture precipitation. One of two varieties are used currently: the Onset RG3-M or the Hydrological Services (HyQuest Solutions Pty Ltd, 2014) TB3. All sites began with either an Onset TBRG or a Texas Electronics TR-525M (R2/R1) but over the years they have been replaced within the 10 km × 10 km network to the configuration documented in Table 1. Currently all sites use a TBRG with a 0.2 mm scale but some earlier TBRG had a 0.1 mm scale. The accuracy for the TB3 is +/- 2% for flow rates of 0-250 mm/hour, and +/- 3% for rates of 250-500 mm/hour (HyQuest Solutions Pty Ltd, 2014); the accuracy of the Onset RG3-M is +/-1% for rates up to 20 mm/hour (Onset Computer Corporation); and the accuracy of the TR-525M-R1 is +/- 1% for rates up to 50 mm/hour (Texas Electronics). Only the TB3 is equipped with a siphon unit which controls the flow of rainfall into the buckets, improving its performance against other TBRG (Devine and Mekis, 2008). Additionally, the filter design of the TB3 is superior in avoiding blockage of the funnel by debris.

Common issues with the TBRG overall include blockage due to debris, mount damage from farm equipment, the occurrence of single tips not related to network-wide rainfall events, and inaccuracy related to hail events. Bird guards were installed on the TB3s where regular debris issues were common. Field calibrations of the TBRG have been completed every two to three years to confirm that the rain gauges were still functioning accurately. If the calibration target was not reached, the TBRG was replaced. A known issue with TBRG-style precipitation gauges is the possibility of single tips due to the retention of water in the bucket or siphon (the latter only in the case of

the TB3). Single tips within the dataset that are not temporally correlated to a rainfall event may not be indicative of rainfall within the 30 minute measurement period. These records have not been removed from the dataset due to the uncertainty in consistently determining validity without removing significant credible data. Another source of error is inaccurate collection of precipitation during hail events, which would then melt and be recorded by the logger.

## 4   Quality Control Process and Data

At the time of publication the network is being run year round, however only May 1 – September 30 is included for each year where shoulder season data exists. The main challenges are difficulties in measurement and calibration of data recorded within the winter and shoulder seasons when the ground is transitioning between a frozen and thawed state (e.g. Williamson et al., 2018). Additionally, TBRGs are not designed for solid
precipitation measurement. Two phases of quality control/quality assurance (QAQC) are performed to warm season data: an automated check and then manual review. The automated phase checks for logger errors and common sensor errors, with the secondary manual review process including a review of field notes and checks of all sensors for known instrument errors and gaps in the automated process. The automatic review begins with the raw measurements and can be completed in near real time, while the secondary manual review is completed on
an as needed basis, or seasonally.

### 4.1   Automated Review Details

The automated review process checks for the limits documented in Table 2 and removes data outside of these thresholds. These checks mainly screen for obvious sensor errors and provide consistency for the next phase of QAQC. Also applied during this process are flags that are using during the manual process to check for common
errors (Table 3).

### 4.2   Manual Review Details

After the automated process, a manual review of the resultant data is conducted to do a final review of the data from each instrument and each site. Hydra Probes are typically reviewed against the site's TBRG, to ensure that jumps in soil moisture correlate with precipitation events. The TBRG are reviewed collectively, as at least for the
sites within the 10 km × 10 km grid precipitation events will be collected by all instruments. This repetition of equipment allows for a relatively high level of confidence in rainfall events and provides useful information to diagnose TBRG collection or measurement errors. Review of field notes and comparison of TBRG between

nearby sites confirms TBRG cleanliness (debris can delay or block rainfall passing into the buckets of the TBRG) and general agreement between sites. When disagreement between a single site and the majority is observed and

confirmed by field visits, the data is removed.

Site visits can potentially cause erroneous data and the data from the day of each site visit is reviewed and edited for (1) extra TBRG tips due to cleaning; (2) erroneous data from the vertically placed 0-5 cm probe when it is moved into and out of the field; (3) other sensor issues that could result in incorrect data (physical damage, disturbance by field equipment or animals); (4) erroneous values from troubleshooting or maintenance checks.

These checks are done in conjunction with review of field notes. Data from each sensor is also visually plotted and reviewed for general operation as sensor malfunction can often be caught in careful review of the sensor parameters; the flags in Table 3 are used at the stage to assist in identifying issues. In this QAQC stage, the focus is on unexplained jumps or drops, gaps, and unusually high or low values that have not yet already been removed during the automated review. Any data diagnosed during this process as erroneous is removed from the final data

set, however as previously mentioned some periods of data that are suspect have been kept in the dataset. The ranges given in Table 3 are only guidelines to assist with manual review: specifically for soil moisture and real dielectric constant, values outside the ranges given may be kept in the data set if the extremes were justified by either the other sensors at the site or the site's TBRG data. The temperature flag is a simple check for frozen ground, as certain years had evidence of frozen ground in May or at the end of September that were removed.

Undoubtedly, certain data issues have been overlooked and new versions of the data will be made as additional QAQC process are developed and implemented.

## 5    Data Availability

The data described here are available at the Federated Research Data Repository (FRDR) (https://dx.doi.org/10.20383/101.0116), as comma-separated-value files. The corresponding author can be

contacted for access to data beyond 2017 as well as any ancillary data.

## 6    Summary

Data from 2007 – 2017, May 1 – Sept 30, from the Kenaston Network in the Brightwater Creek basin in central Saskatchewan, Canada, has been quality controlled and compiled in a standard format. The network consists of two scales of sites, each with 3 – 4 Hydra Probes and a tipping bucket rain gauge. Included in this dataset from

each Hydra Probe is soil moisture, temperature, and real-dielectric constant ($\varepsilon_r$). Some issues with the Hydra Probe have been identified and documented, and the overall network coverage is good. It is anticipated that this dataset

and the data from the network beyond 2017 will continue to provide useful information for remote sensing validation and calibration as well as hydrometeorological modelling efforts.

**Acknowledgements**

The authors would like to thank all of the researchers and summer students who have contributed over the years to the network. This project would not have been possible without the collaboration and good will of these colleagues. Significant among those are Bruce Johnson, Warren Helgason, Dell Bayne, Craig Smith, Anthony Liu, Amber Peterson, Travis Burns, Justin Adams, Matthew Williamson, Sarah Impera, William Woodley, Jon Belanger and Mark Cliffe-Phillips. Funding for the University of Guelph has been provided from the Canadian

Space Agency and grants from the  Natural Science and Engineering Research Council of Canada.

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

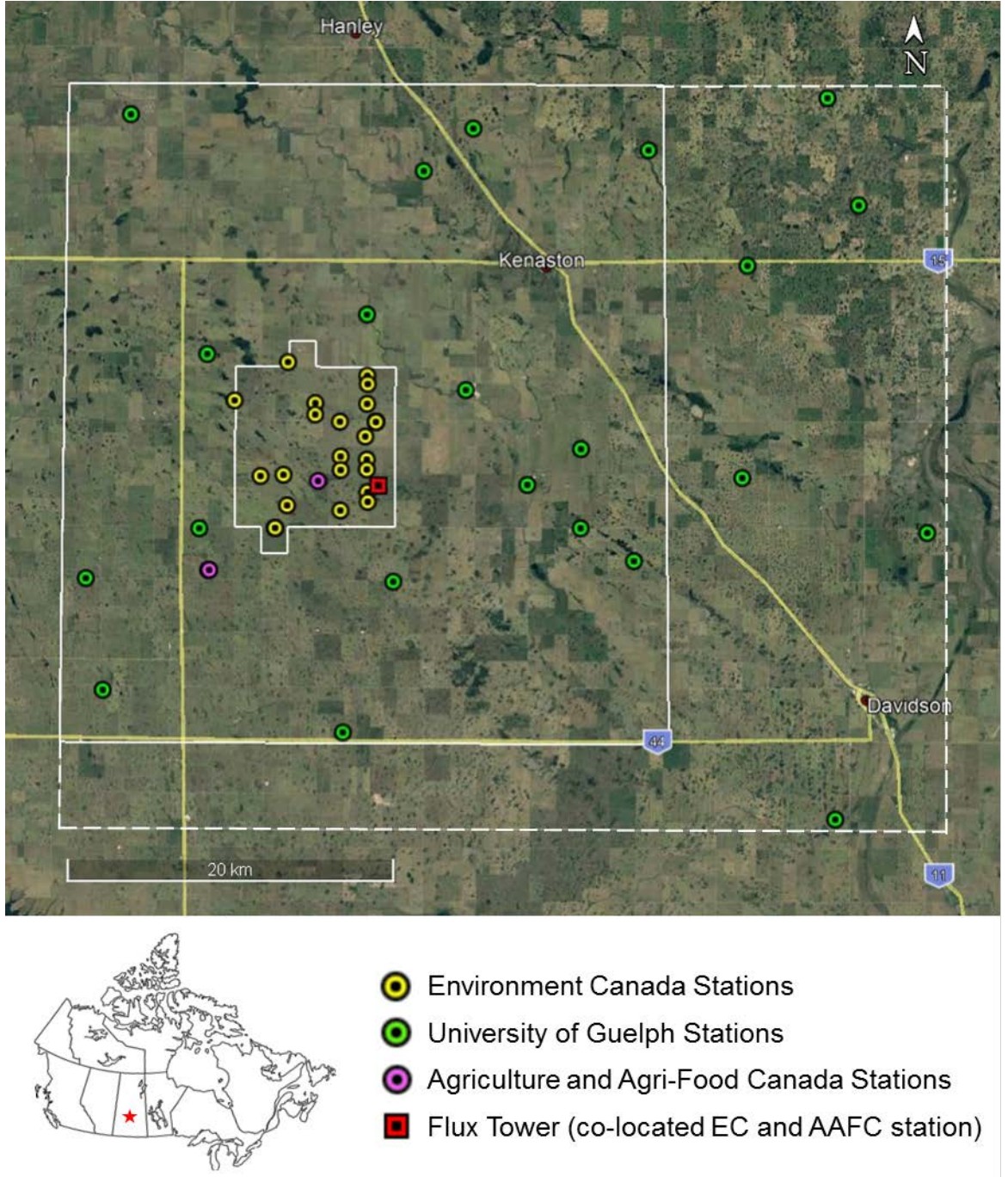

**Figure 1. Map of site locations, the white frames indicating the two scales of the sites. ECCC sites are within a 10 km × 10 km area and University of Guelph sites are within the current 40 km × 40 km area. The dashed line indicates the original larger scale: 45 km by 55 km.**

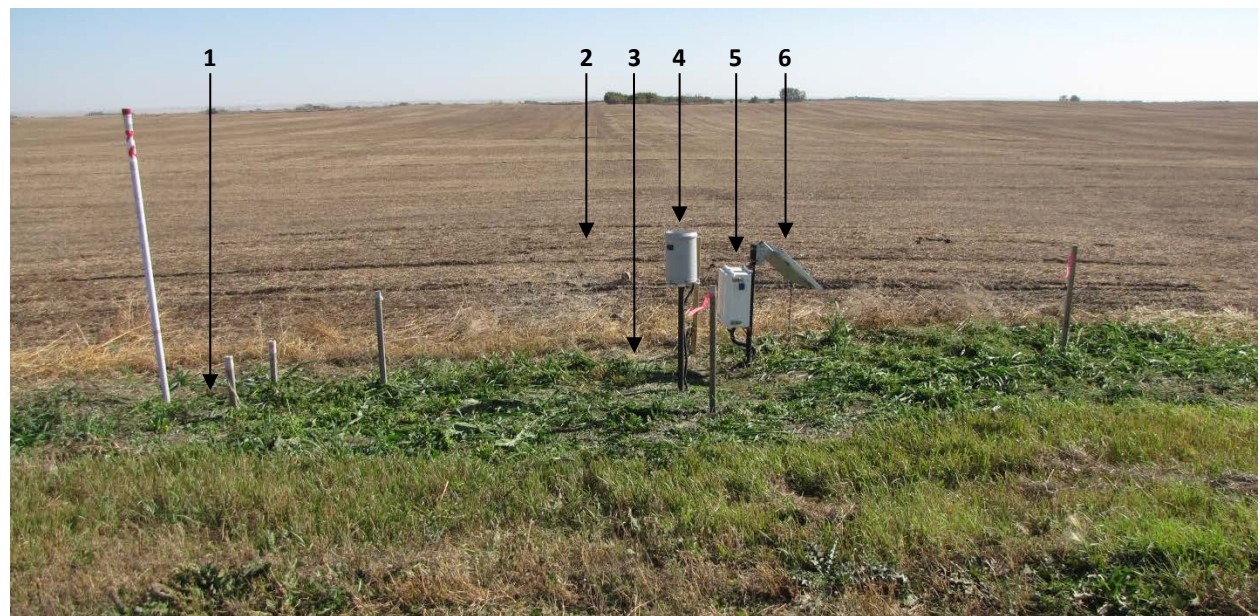

**Figure 2. Typical site installation. The 4-probe sites include at (1) horizontal 5 cm sensor; (2) horizontal 20 and 50 cm sensors and location of vertical 0-5 cm sensor during field season; (3) location of vertical 0-5 cm sensor during off season; (4) tipping bucket rain gauge; (5) loggerbox with datalogger; (6) solar panel. Only ECCC sites have a vertically placed probe. The 3-probe sites are similar, with all probes located at the edge of field at (1).**

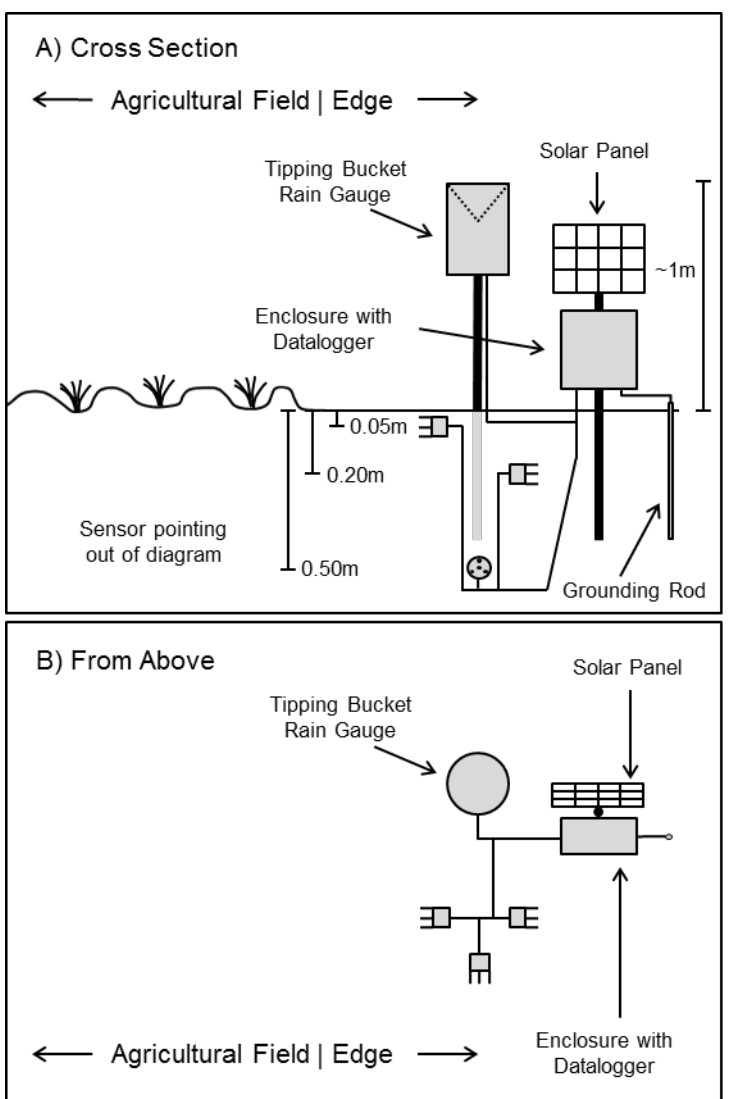

**Figure 3. General configuration of 3-probe soil moisture station.**

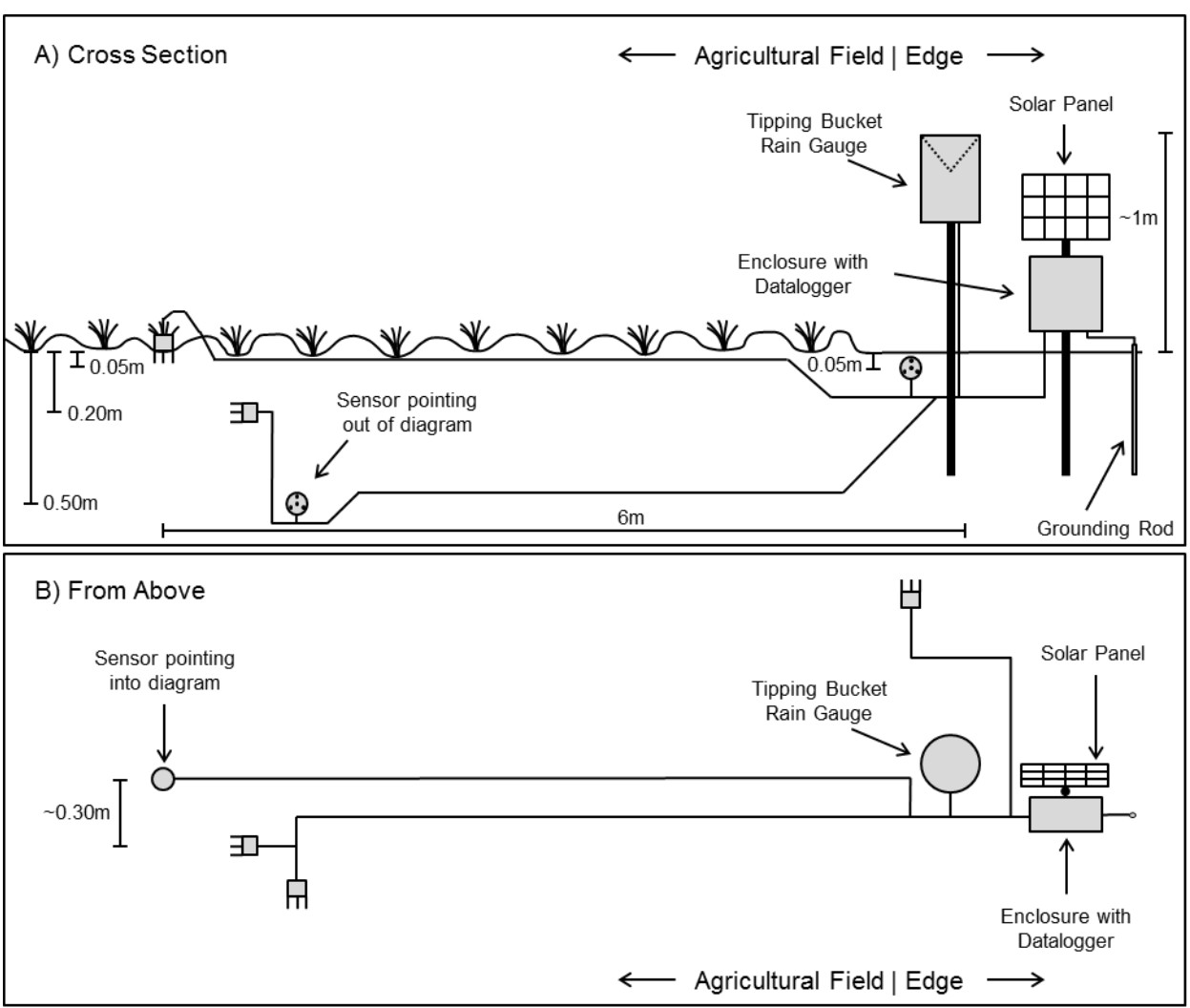

**Figure 4. General configuration of 4-probe soil moisture station.**

**Table 1. Site metadata details including soil texture information.**

| Site ID | Partner | Coordinates | | Instrumentation | | Soil Texture | | | Data Record |
|---------|---------|----------|-----------|----------------|-----------|----------|----------|----------|-------------|
| | | Latitude | Longitude | Hydra Probes | TBRG Type | Sand (%) | Silt (%) | Clay (%) | |
| 2701000 | Guelph | 51.2001 | -106.0156 | 3 | RG3 | 47.1 | 50.3 | 2.6 | 2007-2011 |
| 2701001 | Guelph | 51.5836 | -106.6364 | 3 | RG3 | 33.4 | 63.7 | 2.9 | 2007-2017 |
| 2701002 | Guelph | 51.5767 | -106.3342 | 3 | RG3 | 60.0 | 38.8 | 1.2 | 2007-2017 |
| 2701003 | Guelph | 51.5651 | -106.1799 | 3 | RG3 | 54.7 | 43.0 | 2.3 | 2007-2011 |
| 2701004 | Guelph | 51.5914 | -106.0146 | 3 | RG3 | 54.7 | 42.9 | 2.2 | 2007-2010 |
| 2701005 | Guelph | 51.4529 | -106.5672 | 3 | RG3 | 35.7 | 60.8 | 3.5 | 2007-2017 |
| 2701006 | Guelph | 51.5534 | -106.3776 | 3 | RG3 | 58.4 | 40.3 | 1.3 | 2007-2017 |
| 2701007 | Guelph | 51.5021 | -106.0927 | 3 | RG3 | 61.7 | 37.0 | 1.3 | 2007-2011 |
| 2701008 | Guelph | 51.5351 | -105.9950 | 3 | RG3 | - | - | - | 2007-2011 |
| 2701009 | Guelph | 51.3300 | -106.6724 | 3 | RG3 | 31.0 | 52.0 | 17.0 | 2007-2015 |
| 2701010 | Guelph | 51.4374 | -106.2222 | 3 | RG3 | 47.1 | 50.3 | 2.6 | 2007-2009 |
| 2701011 | Guelph | 51.3864 | -106.0971 | 3 | RG3 | 34.5 | 62.6 | 2.9 | 2007-2010 |
| 2701012 | Guelph | 51.3564 | -105.9351 | 3 | RG3 | 23.8 | 72.4 | 3.8 | 2007-2010 |
| 2701013 | Guelph | 51.2690 | -106.6568 | 3 | RG3 | 30.0 | 49.0 | 21.0 | 2007-2017 |
| 2701014 | Guelph | 51.2468 | -106.4460 | 3 | RG3 | 25.0 | 54.0 | 21.0 | 2007-2017 |
| 2701015 | Guelph | 51.3577 | -106.5729 | 3 | RG3 | 28.0 | 47.0 | 25.0 | 2007-2017 |
| 2701016 | Guelph | 51.4020 | -106.2385 | 3 | RG3 | 39.8 | 52.2 | 8.0 | 2014-2017 |
| 2701017 | Guelph | 51.4749 | -106.4268 | 3 | RG3 | 10.6 | 48.3 | 41.1 | 2014-2017 |
| 2701018 | Guelph | 51.3292 | -106.4025 | 3 | RG3 | 10.5 | 63.7 | 25.9 | 2014-2017 |
| 2701019 | Guelph | 51.3824 | -106.2853 | 3 | RG3 | 39.0 | 31.2 | 29.8 | 2014-2017 |
| 2701020 | Guelph | 51.3588 | -106.2386 | 3 | RG3 | 33.6 | 60.6 | 5.8 | 2014-2017 |
| 2701021 | Guelph | 51.3409 | -106.1918 | 3 | RG3 | 54.5 | 34.1 | 11.4 | 2014-2017 |
| 2701022 | ECCC | 51.3817 | -106.4159 | 4 | TB3 | 26.2 | 60.5 | 13.3 | 2007-2017 |
| 2701023 | ECCC | 51.3679 | -106.4492 | 4 | TB3 | 37.0 | 41.0 | 22.0 | 2007-2017 |
| 2701024 | ECCC | 51.3706 | -106.4960 | 4 | TB3 | 34.0 | 50.0 | 16.0 | 2007-2017 |
| 2701025 | ECCC | 51.4488 | -106.4960 | 4 | TB3 | 25.4 | 56.3 | 18.2 | 2007-2017 |
| 2701026 | ECCC | 51.3727 | -106.4253 | 4 | TB3 | 28.6 | 57.3 | 14.1 | 2007-2017 |
| 2701027 | ECCC | 51.3780 | -106.4256 | 4 | TB3 | 28.0 | 59.0 | 13.0 | 2007-2017 |
| 2701028 | ECCC | 51.3872 | -106.4994 | 4 | TB3 | 42.0 | 41.0 | 17.0 | 2007-2017 |
| 2701029 | ECCC | 51.3865 | -106.5195 | 4 | TB3 | 39.0 | 44.0 | 17.0 | 2007-2017 |
| 2701030 | ECCC | 51.3958 | -106.4262 | 4 | TB3 | 31.0 | 46.0 | 23.0 | 2007-2017 |
| 2701031 | ECCC | 51.3974 | -106.4493 | 4 | TB3 | 26.6 | 55.7 | 17.7 | 2007-2017 |
| 2701032 | ECCC | 51.3904 | -106.4262 | 4 | TB3 | 15.7 | 52.0 | 32.3 | 2007-2017 |
| 2701033 | ECCC | 51.3900 | -106.4492 | 4 | TB3 | 26.0 | 50.0 | 24.0 | 2007-2017 |

| | | | | | | | | | |
|---|---|---|---|---|---|---|---|---|---|
| 2701034 | ECCC | 51.4164 | -106.4184 | 4 | TB3 | 29.0 | 49.0 | 22.0 | 2007-2017 |
| 2701035 | ECCC | 51.4164 | -106.4501 | 4 | TB3 | 26.0 | 51.0 | 23.0 | 2007-2017 |
| 2701036 | ECCC | 51.4084 | -106.4277 | 4 | TB3 | 33.0 | 46.0 | 21.0 | 2007-2011 |
| 2701037 | ECCC | 51.4262 | -106.4262 | 4 | TB3 | 26.8 | 51.4 | 21.8 | 2007-2017 |
| 2701038 | ECCC | 51.4265 | -106.4718 | 4 | TB3 | 13.8 | 57.0 | 29.2 | 2007-2017 |
| 2701039 | ECCC | 51.4202 | -106.4718 | 4 | TB3 | 30.2 | 51.3 | 18.5 | 2007-2017 |
| 2701040 | ECCC | 51.4277 | -106.5428 | 4 | TB3 | 31.8 | 46.1 | 22.1 | 2007-2017 |
| 2701041 | ECCC | 51.4166 | -106.4184 | 4 | TB3 | 20.0 | 43.0 | 37.0 | 2007-2017 |
| 2701042 | ECCC | 51.4370 | -106.4258 | 4 | TB3 | 12.7 | 70.1 | 17.2 | 2007-2017 |
| 2701043 | ECCC | 51.3582 | -106.5064 | 4 | TB3 | 50.0 | 32.0 | 18.0 | 2007-2017 |
| 2701044 | ECCC | 51.4416 | -106.4262 | 4 | TB3 | 24.6 | 59.5 | 15.9 | 2007-2017 |

[a] TBRG types: Onset RG3 and Hydrological Services TB3.

**Table 2. Limits applied in QC1 – data removed**

| Parameter | Limits |
|---|---|
| Temperature (°C) | $-60 < x < 60$ |
| Real dielectric constant ($\varepsilon_r$, unit-less) | $0 < x < 90$ |
| Soil moisture, loam calibration (VWC, ($m^3m^{-3}$)) | $0 < x < 1.0$ |

**Table 3. QAQC flags for manual review**

| Parameter | QAQC Checks |
|---|---|
| Temperature (°C) | $x < 0$ |
| Real dielectric constant ($\varepsilon_r$, unit-less) | $x < 2.4$ |
| Soil moisture, loam calibration (VWC, ($m^3m^{-3}$)) | $0.02 < x < 0.6$ |