# Peer review of "An 11-yr (2007 – 2017) soil moisture and precipitation dataset from the Kenaston Network in the Brightwater Creek basin, Saskatchewan, Canada."

_Earth System Science Data, 2018_

## Referee Comment (RC1) · Anonymous Referee #1 · 22 Nov 2018

The manuscript refers to a 11 year dataset of soil moisture, soil temperature and precipitation from more than 50 sites within a 1600 km2 region collected during the growing season for an agricultural prairie landscape.

GENERAL COMMENTS The manuscript and dataset are of value to those involved in soil moisture research. Aside from small errors the manuscript is of appropriate length and reasonably clear. The data set is well organized and consistently structured. The data is unique and very useful, especially as the dielectric value and soil temperature is provided, and its coverage of 11 years. The dataset is presented in a usable format (checked with R and Excel). Sufficient background is given on sensors and usage.

[Figure]

Before publication the manuscript requires numerous, but minor, improvements (see Specific Comments), and the data set requires further quality improvements.

The dataset presented is limited to May thru Sept although it appears that year-round data is available. Although the dielectric value of soils is strongly affected by freezing it can still be of value to researchers, especially if soil temperature of the probe is given. For example, see Kelleners and Norton, 2012 (Soil Science of America) and Roy et al 2017 (Remote Sensing of Environment). Both studies used Hydra probes. If available it is recommended that the entire year-round data set be made available, with appropriate caveats given about freezing conditions.

Dataset needs further quality control: high moisture content values greater than 0.60 m3/m3 are present and in one known case greater than 1.0 m3/m3; moisture content fluctuations between 30 minute intervals for many probes (especially at 20 and 50 cm) are greater then acceptable (>0.02 m3/m3 and up to 0.10 m3/m3). Although the fluctuations are stated as being caused by possible salinity it was not made clear by the authors whether they were to be kept or removed from the data set. They should be kept and clarification (and caveat) statements need to be added. Additionally, some data sets do not extend up to the years indicated in Table 1 (e.g. they only go as far as 2010, not to 2013).

There is confusion around the terms 'two spatial scales' and 10 km2 and 40 km2. Spatial scale implies different spacings or densities. The areas measure 10 km by 10 km and 40 km by 40 km thus are more 100 km2 by 1600 km2 in area. Perhaps referring to density of the stations per km2 or their average spacing would provide more information for the reader. The authors refer to the manufacturers loam setting being used to calculate the moisture contents, however this 'loam' equation cannot be found in the provided references.

SPECIFIC COMMENTS/QUESTIONS 3:67-70, confusion with sensor locations; make it clear in the text and in Figs 2 and 3 which probes are set within the tilled field and
which in the field edge. Lines 67 to 68 (5 cm, 20 cm, and 50 cm) do not state location, however Figures 2 and 3 indicate the 20 cm and 50 cm are within the tilled field. Also Figure 2 states "(3) location of vertical 0-5 cm sensor during off season". This implies the sensor is still active and recording or it is 'off'? Clearly indicate this in text. 4:85, give length of tines 4:102, why not publish the calibration equations in the paper or on the data web site? What is the degree of moisture difference between the calibrated probes and that given by the manufacturer? 5:118, some confusion about that of electrical conductivity (EC) measurements. Do Hydra probes measure EC? If so, was this measured by the installed probes? If it was measured, then it should be mentioned in the text and rational given as to why it is not published – as it appears it might be useful in discerning problem readings which are in the data set (e.g. site 2701023 50 cm probe). 6: 149-150, and Table 3 gives a flag for soil moisture not being greater than 0.6 m3/m3, however many data sets have values greater then this at various depths (e.g., 2701023 at 50 cm). Were these to have been removed? If so then please check all data sets. If they are to stay in then clearly indicate so. 6:155, the manuscript does not indicate which set of sites were the 'dense set'. When were the soil moisture and precipitation sites established? This should be stated in the paper and not in the Summary. Are they still maintained and visited? How often were the sites visited? The document states 'regularly' however is this once a year or 5 times? Table 1 shows Data records up to 2017 – does this mean the data was not collected after 2017 or is this when the table was compiled? Figures 2 and 3; make it clear that only the ECCC sites (the dense network) have the vertical 5 cm sensor in the agricultural field. As indicated by the authors some of the data is more variable than expected (e.g. possible saline conditions at 20 and 50 cm depth, see lines 119-122). Although it is stated on lines 168 to 169 that erroneous data is removed from the final data set there are numerous instances of 'unexplained drops and unusually high or low values'. If some of the values retained are due to possible saline conditions then it should be clarified that these values were kept but the user must be careful about their interpretation. See Technical Corrections for examples.

TECHNICAL CORRECTIONS 1:21, insert 'the' before 'hydrological cycle'. For clarification change the following: "While soil moisture constitutes a small portion of the global water cycle, it has a...". 2:43, As this is an international journal add the following "..a typical prairie agricultural..." to help define 'typical'. 2:46, 'considered'. 2:46, remove 'in general'. 2:47, incomplete sentence – suggest the following "Texture of the soils in the region is predominantly silt loam but ranges from sandy loam to clay.". 2:49, remove 'over the years'. 2:53-55, Do the references given in line 55 refer to the 2010 (CanEx-SM10) study? If not then perhaps create two sentences. Remove 'previous' as not necessary.. 3:57-58, clearly state that the AAFC stations are not included in this data set. 3:67, add an 's'; "Additionally, sites at ...". 3:68, insert 'the' before "site at the...". 3:71, "..vertically placed probe,..." add the 'd' to place. 3:73-74, "the sum over the '30 minute' interval for the TBRG.". 3:76, '..within the Kenaston network, ...". 3:79, "and to check for ". 3:79, "Sites with a vertically...". 4:84, there is no Stevens Water Monitoring Systems Inc 2009 or Burns 2016 in the Reference list. Why the double parenthesis – is Burns 2016 a reference within the other? 4: 99, Burns et al 2014 is missing in the reference list (or year not given). 4:102, always provide a space between the value and the units; e.g. "5 cm". Check through the manuscript for this. 4:117, what is a 'timestamp' ? Is this the 30 minute interval? 4:137, replace 'currently' with a date or at least a year as the article can be in existence for much longer than the network. 5:130, why 'regularly' completed? Were calibrations required every year or so or just once? What about the RG3's – did they require calibration? 5:137, reword removing 'at maximum' as this is too confusing when referring to dates. 8:195, a year is needed for the first Burns et al reference. Table 1, should indicate which site is ECCC and which is Guelph (using same terminology in Figure 1). Table 1, Note at end, "Onset RG3 and ..." insert space. Table 3, if Conductivity refers to soil Electrical Conductivity it should have units, e.g. dS/m

Dataset web pages: 'moisture' is repeatedly spelt wrong in the 'Readme' file. Below are some of the issues found with the data sets. Not all data sets were investigated. V2701000 for 2007; H5 cm probe varies by 0.02 up to 0.1 m3/m3 each time interval so

it appears that there could be a choice of three possible sets of data to choose from each day. This type of fluctuation appears to be common for most probes with the range of fluctuation becoming greater at certain moisture contents and more so at 50 cm depth. Could be a function of both the Hydra probe and salinity?? 20 cm probe has values greater than 1.0, (July 2007) likely because the dielectric values are greater than 100. 50 cm probe has values vary by more than 0.1 m3/m3 within each day. V2701001 and V2701002 something with the dates that R did not like. V2701003 had no data from 2011 on (Table 1 states data from 2007-2013). V2701004, 50 cm VWC varies too much and no data present from 2011 on. V2701005 data appears reasonable in values and range. No data present from 2010 on. V2701006 data appears reasonable in values and range. No data present from 2011 on. V2701023, 50 cm probe has high range of daily fluctuations and values greater than 0.60 m3/m3. V2701025, 50 cm depth has a strange fluctuation that when plotted over the season it appears to have three distinct sets of data. This is not seen for the other sensors of this site. Many dielectric values are high and some sensors show very little response to seasonal rains or drying events. V2701034 has values > 0.60 at the 20 cm depth. At 50 cm depth moisture readings indicate saturation (0.50 or higher). V2701035 has values > 0.60 at the 50 cm depth

---

## Referee Comment (RC2) · Anonymous Referee #2 · 9 Dec 2018

An 11-yr (2007–2017) soil moisture and precipitation dataset from the Kenaston Network in the Brightwater Creek basin, Saskatchewan, Canada

General Comments The authors describe and present an alternative soil moisture, precipitation and temperature dataset based on in situ observation to calibrate and validate remote sensing measurements and hydrological model outputs. The data covers the period of 2007-2017 at two domains with different station density in a hydrometeorological network situated within the Brightwater Creek basin, east of Kenaston, Saskatchewan.

Specific Comments: This article is well written and well organized. These data are

clearly described and well formatted (Excel). The article targets an important issue of calibration and validation of growing space-based observations and hydrological model outputs against ground-based measurements. It is crucial to evaluate the reliability of those products before routine use at a global scale. However, there are a few points in the article that can benefit from improvement: - 75: It would be clearer if the loam calibration equation were included. - It is not clear why the authors have chosen only 11 years and whether this dataset will be continued or the operations has stopped after 2017 - If the stations are actively reporting the measurements, will the dataset be publicly available later? In addition, how long does it take data to become publicly available after ingest? - Since the network was designed for validation purposes, a comparison between the quality controlled data and existing datasets like SMAP and SMOS could be beneficial. - 94: Include the equation and the reference in this section - 99-101: Then what is the range of the uncertainty involved in these calculations? - 102: Providing these equations in the text will make it easier for the readers to follow your method. - 104-112: The issue is explained very well, but the authors do not clarify whether they have removed such problematic measurements from the data or if they are just recorded as they are. - The sources of errors in the dataset are explained, but the study will benefit from a calculated estimate of such errors.

Technical Corrections: 1. 12: Change ESA to European Space Agency. 2. 14-15: According to Fig. 1's scale, the two domains are 10x10 (100km2) and 40x40 (1600 km2) 3. 14-15: Please clarify the wording, because it is not clear if this is describing two different domains or two different domains with different spatial resolution among the sensors. The wording is not clear but the figure clearly shows an outer domain and a higher-resolution inner domain. 4. 35-36: The last sentence need more explanation: "The high resolution of the network sites allows for both intergrid and intragrid valida- tion". 5. 59-61: The author clearly states that AAFC stations are located within the pasture sections but it is not clear what type of landscape the ECCC and University of Guelph cover. Please clarify this in the text. 6. 63: Please refer to comment #2 7. 64: "45 x 55 km" should change to "45 x 55 km2 " and also x should be replaced my

multiplication symbol 8. 68: please refer to comment#2. Also there is a typo in km2- 9. 74: "is" should be replaced by "are" 10. 79: "regularly" should be explained in detail. How often are the sites visited? 11. 80: "more frequently". Please refer to comment# 10 12. 99: adding a comma after "manufacturer supplied" will make the sentence more clear 13. 126: "10 km2" should be replaced by "10 km" 14. 137: keep the consistency between the used words: year-round (line 40) 15. 138: "occur" should be replaced by "occurring" 16. 140: How do the thunderstorms producing solid precipitation (e.g. hailstones) in the growing season will add to the error of your measurements? 17. 47-50: what is the source of these thresholds? Please add a reference. 18. 154: What about irrigation. The abstract mentioned that the site is an agricultural site with croplands but irrigation is not mentioned anywhere in the text. 19. 182-186: The external funding sources for these operations should be mentioned in the acknowledgement. 20. 226-231: The references are in alphabetical and chronological orders. Rowlandson et al. 2013 should precede Rowlandson et al. 2015 21. 258-260: please refer to comment # 2

---

## Referee Comment (RC3) · Anonymous Referee #3 · 18 Dec 2018

This paper presents a soil moisture and precipitation dataset from a hydro-meteorological network located in the Canadian Prairies (Saskatchewan). The paper is well written and the data are available on a Canadian platform designed to host research data. Therefore, I recommend the publication of this paper in ESSD subject to minor revisions outlined below.

- Specific comments

Introduction: the introduction is rather short and some elements could be added to better insist on the need of detailed hydrometeorological dataset in this region of Canada. For example, at P2 L 28-30, it would be interesting to add a few sentences and references on the remote sensing of soil moisture and the associated challenges, including the need for calibration at reference sites. The second paragraph could be also more accurate:

- What is the "unique combination of landscape and climatic conditions" mentioned at P 2 L 33-34?

- What are the other few existing monitoring networks available in the Canadian Prairies? Over the last few years, these networks have been used to evaluate land surface models applied for hydrological and weather forecasting in Canada (Garnaud et al., 2016,2017).

Garnaud, C., S. Bélair, A. Berg, and T. Rowlandson, 2016: Hyperresolution Land Surface Modeling in the Context of SMAP Cal–Val. J. Hydrometeor., 17, 345–352,https://doi.org/10.1175/JHM-D-15-0070.1

Garnaud, C., S. Bélair, M.L. Carrera, H. McNairn, and A. Pacheco, 2017: Field-Scale Spatial Variability of Soil Moisture and L-Band Brightness Temperature from Land Surface Modeling. J. Hydrometeor., 18, 573–589, https://doi.org/10.1175/JHM-D-16-0131.1

P 2 L 40-41: the authors mention the presence of an eddy-covariance tower (also mentioned in the abstract). No additional information are available in the rest of the text. Are the data of this tower available from one of the institutional partner involved in this community site? If not, can the author add this dataset to their paper and make this dataset available on the FRDR website? It would be extremely valuable for the evaluation of land surface and hydrological models.

P 3 L 56: how does the University of Saskatchewan contribute to the community site?

P 3 L 57-58: the AAFC stations are of potential interest for any studies in this area. I recommend the author to mention in the text the number of AAFC stations located in the area and to show their location on Figure 1.

P 3 L 63: the spatial scales are not accurate when compared to Fig. 1. Are the authors mentioning an area of 10*10 km2 instead of 10 km2? And 40*40 km2 instead of 40 km2? The abstract should be modified accordingly.

P 3 L 63-65: it would be interesting for the readers to know the number of stations in the network and to refer to Table 1 to mention that the exact position of each stations is given in Table 1.

P 3 L 75-78: which institutional partner has collected the additional data? Can these data be obtained on request?

P 3 L 79-81: what is the typical frequency of the visits at the sites in summer time?

P 4 L 100: the loam calibration equation should be given in the paper since it is a paper focusing on the data.

P 4 L 101-102: what is the impact of using in-situ calibration equations on the accuracy of the computation of soil moisture? How large is the decrease in accuracy when using the loam calibration equation? Even if the in-situ calibration equations are not available for each probe, this comparison would be very useful for the reader to better understand the accuracy of the dataset presented in this paper.

P 4 L 103-109: which treatment is applied to the soil moisture data when measurements issues occur with the Hydra Probe? Based on Table 3, it seems that these issues are identified during the automatic QC but Section 4.2 does not detail the final treatment applied to the soil moisture data.

P 5 L 125-126: what is the impact of the replacement of the rain gauges at the ECCC sites on the quality and the consistency of the times series of precipitation?

P 6 L 149-150: it is not clear in Section 4.2 how the flags described in Table 3 are used during the manual review process. I recommend the authors to describe the specific treatment applied on the dataset for each flag.

[Figure]

P 7 L 172: mention the format of the data on the FRDR website,

P 7 L 179-180: is the data acquisition still continuing at the ECCC stations and at the University of Guelph stations? If it is the case, will the data be made available in a near future? At which frequency and on which platform? The fact that the data acquisition is still continuing is really important and should be clearly mentioned in the conclusion and also in the introduction.

P 13: do the stations equipped with 4 probes share the same configuration? In particular, are the probes at 20 cm and 50 cm always located in the ground below the agricultural field as shown on Fig. 3?

P 14: Table 1: it would be interesting to know in the table which stations belong to ECCC and which stations belong to the University of Guelph.

- Technical Comments

Abstract L 9: mention that Saskatchewan is located in Canada.

P 4 L 84: the references are not correct.

- Comments on the dataset

Metadata: the metadata on the FRDR website does not contain the location of each stations. These information are only given in Table 1. of the submitted paper. I recommend the authors to add an ascii file containing the locations of each station. This file could have the same content as Table 1 and could be used with Python or R by a person interested in this dataset.

---

## Author Comment (AC1) · 5 Mar 2019

Please see the attached .pdf for our responses to the reviewers. We thank all three for their constructive comments. Each has been addressed and we hope that our responses are adequate.

The data has also been updated in the FRDR repository, as per QAQC suggestions by the reviewers.

Please also note the supplement to this comment:

[Figure]

https://www.earth-syst-sci-data-discuss.net/essd-2018-122/essd-2018-122-AC1-supplement.zip

---

## Author Response (AR1)

GENERAL COMMENTS The manuscript and dataset are of value to those involved in soil moisture research. Aside from small errors the manuscript is of appropriate length and reasonably clear. The data set is well organized and consistently structured. The data is unique and very useful, especially as the dielectric value and soil temperature is provided, and its coverage of 11 years. The dataset is presented in a usable format (checked with R and Excel). Sufficient background is given on sensors and usage.

Before publication the manuscript requires numerous, but minor, improvements (see Specific Comments), and the data set requires further quality improvements.

The dataset presented is limited to May thru Sept although it appears that year-round data is available. Although the dielectric value of soils is strongly affected by freezing it can still be of value to researchers, especially if soil temperature of the probe is given. For example, see Kelleners and Norton, 2012 (Soil Science of America) and Roy et al 2017 (Remote Sensing of Environment). Both studies used Hydra probes. If available it is recommended that the entire year-round data set be made available, with appropriate caveats given about freezing conditions.

**While other studies have included winter and shoulder season data in their analyses, data from these periods require significantly more QAQC, with care taken to identify when the sensors are in frozen, partially frozen, and thawed ground. The two major purposes of this dataset, remote sensing and hydrological model validation, are typically not yet capable of doing that interpretation themselves and require straightforward inputs. Progress is being made to create repeatable, automated processes that will allow careful identification of freezing and thawing periods to avoid biases and inaccurate identification of frozen or thawed ground, at which point a new version of the database will be released.**

Dataset needs further quality control:

high moisture content values greater than 0.60 m3/m3 are present and in one known case greater than 1.0 m3/m3;

**The data has been more carefully reviewed and this issue has been corrected.**

moisture content fluctuations between 30 minute intervals for many probes (especially at 20 and 50 cm) are greater then acceptable (>0.02 m3/m3 and up to 0.10 m3/m3).

**Fluctuations of this type have been more fully described in the paper. With an error tolerance on the Hydra Probes of +/- 0.03, some amount of fluctuation could be reasonable. Some errors of this type have been removed from the dataset, however some caution is required by data users.**

Although the fluctuations are stated as being caused by possible salinity it was not made clear by the authors whether they were to be kept or removed from the data set. They should be kept and clarification (and caveat) statements need to be added.

**Further clarification on the data fluctuations has been given and a statement added that data users will need to be aware of this issue in the dataset as not all periods of significant fluctuation have been removed.**

Additionally, some data sets do not extend up to the years indicated in Table 1 (e.g. they only go as far as 2010, not to 2013).

**Corrected.**

There is confusion around the terms 'two spatial scales' and 10 km2 and 40 km2. Spatial scale implies different spacings or densities. The areas measure 10 km by 10 km and 40 km by 40 km thus are more 100 km2 by 1600 km2 in area. Perhaps referring to density of the stations per km2 or their average spacing would provide more information for the reader.

**Corrected.**

The authors refer to the manufacturers loam setting being used to calculate the moisture contents, however this 'loam' equation cannot be found in the provided references.

**Equation has now been included and confirmed in the included reference: The equation [A2] is listed in Appendix C and the coefficients are noted in Section 5.2.3 - Soil Moisture Calibrations.**

SPECIFIC COMMENTS/QUESTIONS

3:67-70, confusion with sensor locations; make it clear in the text and in Figs 2 and 3 which probes are set within the tilled field and which in the field edge.

**An additional figure has been added and more detail has been given about sensor location at each type of site.**

Lines 67 to 68 (5 cm, 20 cm, and 50 cm) do not state location, however Figures 2 and 3 indicate the 20 cm and 50 cm are within the tilled field. Also Figure 2 states "(3) location of vertical 0-5 cm sensor during off season". This implies the sensor is still active and recording or it is 'off'? Clearly indicate this in text.

**An additional figure has been included to clarify the location of the probes at each type of site and text has been added to the Soil Moisture and Precipitation Site Details section.**

4:85, give length of tines

**Added.**

4:102, why not publish the calibration equations in the paper or on the data web site? What is the degree of moisture difference between the calibrated probes and that given by the manufacturer?

**Further details were provided in this section giving RMSE values found by a previous study and the continued issues in calibration method. These references go into much greater detail regarding the need for calibration equations beyond those given by the manufacturer.**

5:118, some confusion about that of electrical conductivity (EC) measurements. Do Hydra probes measure EC? If so, was this measured by the installed probes? If it was measured, then it should be mentioned in the text and rational given as to why it is not published – as it appears it might be useful in discerning problem readings which are in the data set (e.g. site 2701023 50 cm probe).

**Hydra Probes are capable of measuring electrical conductivity however not all sites collect this data and it has not been measured for the entire data record, which is why it is not included. Statements have been added to clarify that some periods of high fluctuation have been removed, but not all, and data users will need to review this themselves.**

6: 149-150, and Table 3 gives a flag for soil moisture not being greater than 0.6 m3/m3, however many data sets have values greater then this at various depths (e.g., 2701023 at 50 cm). Were these to have been removed? If so then please check all data sets. If they are to stay in then clearly indicate so.

**The flags given, which include the test of soil moisture greater than 0.6 m3/m3, are not meant to specifically indicate incorrect values, but data intervals that require extra attention. In certain conditions soil moisture greater than 0.6 m3/m3 have been recorded at the sites that even after investigation appear to be correct. While the porosity of loam soils is typically limited to 60%, the inclusion of organics during the growing season can increase this upper limit. A statement has been added to the Manual Review Details section clarifying the use of the flags and the data has been further reviewed removing missed issues or confirming previous decisions to include or remove data.**

6:155, the manuscript does not indicate which set of sites were the 'dense set'.

**Rephrased.**

When were the soil moisture and precipitation sites established? This should be stated in the paper and not in the Summary.

**Added in the network description section.**

Are they still maintained and visited?

**Sentence added in the network description section clarifying that as of publication a majority of the sites are still active.**

How often were the sites visited? The document states 'regularly' however is this once a year or 5 times?

**Sentence added giving clarification of the frequency of site visits.**

Table 1 shows Data records up to 2017 – does this mean the data was not collected after 2017 or is this when the table was compiled?

**As per other comments, details have been added in various parts of the paper indicating that the sites are (as of the time of publication) still in operation and additional data can be requested.**

Figures 2 and 3; make it clear that only the ECCC sites (the dense network) have the vertical 5 cm sensor in the agricultural field.

**Sentence added to each figure caption.**

As indicated by the authors some of the data is more variable than expected (e.g. possible saline conditions at 20 and 50 cm depth, see lines 119-122). Although it is stated on lines 168 to 169 that erroneous data is removed from the final data set there are numerous instances of 'unexplained drops and unusually high or low values'. If some of the values retained are due to possible saline conditions then it should be clarified that these values were kept but the user must be careful about their interpretation. See Technical Corrections for examples.

**Additional data review has been completed. Sentences have been included to clarify that some periods of high variability remain in the dataset and users need to be cautious. Certain other drops and jumps are similar in that they have been left in the dataset and caution may be required.**

TECHNICAL CORRECTIONS

1:21, insert 'the' before 'hydrological cycle'. For clarification change the following: "While soil moisture constitutes a small portion of the global water cycle, it has a. . .".

**Fixed.**

2:43, As this is an international journal add the following "..a typical prairie agricultural. . ." to help define 'typical'.

**Fixed.**

2:46, 'considered'.

**Fixed.**

2:46, remove 'in general'.

**Corrected to 'typically' as these non-contributing areas can, in certain circumstances, contribute to streamflow.**

2:47, incomplete sentence – suggest the following "Texture of the soils in the region is predominantly silt loam but ranges from sandy loam to clay.".

**Agreed, fixed.**

2:49, remove 'over the years'.

**Fixed.**

2:53-55, Do the references given in line 55 refer to the 2010 (CanEx-SM10) study? If not then perhaps create two sentences. Remove 'previous' as not necessary.

**Corrections made with separate sentences as suggested.**

3:57-58, clearly state that the AAFC stations are not included in this data set.

**Clarification given in at the end of the section.**

3:67, add an 's'; "Additionally, sites at . . .".

**Fixed.**

3:68, insert 'the' before "site at the. . .".

**A repeat of the previous comment, fixed.**

3:71, "..vertically placed probe,. . ." add the 'd' to place.

**Fixed.**

3:73-74, "the sum over the '30 minute' interval for the TBRG.".

**Fixed.**

3:76, '..within the Kenaston network, . . .".

**Fixed.**

3:79, "and to check for ".

**Fixed.**

3:79, "Sites with a vertically. . .".

**Fixed.**

4:84, there is no Stevens Water Monitoring Systems Inc 2009 or Burns 2016 in the Reference list. Why the double parenthesis – is Burns 2016 a reference within the other?

**Corrected.**

4: 99, Burns et al 2014 is missing in the reference list (or year not given).

**Year added to existing reference.**

4:102, always provide a space between the value and the units; e.g. "5 cm". Check through the manuscript for this.

**Fixed and all other instances checked.**

4:117, what is a 'timestamp' ? Is this the 30 minute interval?

**Clarified: "typical variation between successive measurement intervals (timestamps)"**

4:137, replace 'currently' with a date or at least a year as the article can be in existence for much longer than the network.

**Rephrased.**

5:130, why 'regularly' completed? Were calibrations required every year or so or just once? What about the RG3's – did they require calibration?

**Clarification of the calibration process was included.**

5:137, reword removing 'at maximum' as this is too confusing when referring to dates.

**Rephrased.**

8:195, a year is needed for the first Burns et al reference.

**Added.**

Table 1, should indicate which site is ECCC and which is Guelph (using same terminology in Figure 1).

**Table 1 has been updated to include these details.**

Table 1, Note at end, "Onset RG3 and . . ." insert space.

**Fixed.**

Table 3, if Conductivity refers to soil Electrical Conductivity it should have units, e.g. dS/m

**Fixed.**

Dataset web pages:

'moisture' is repeatedly spelt wrong in the 'Readme' file.

**Corrected.**

Below are some of the issues found with the data sets. Not all data sets were investigated.

V2701000 for 2007; H5 cm probe varies by 0.02 up to 0.1 m3/m3 each time interval so it appears that there could be a choice of three possible sets of data to choose from each day. This type of fluctuation appears to be common for most probes with the range of fluctuation becoming greater at certain moisture contents and more so at 50 cm depth. Could be a function of both the Hydra probe and salinity?? 20 cm probe has values greater than 1.0, (July 2007) likely because the dielectric values are greater than 100. 50 cm probe has values vary by more than 0.1 m3/m3 within each day.

**Multiple other comments are similar and statements have been given here as well as in the paper clarifying the potential for salinity issues and that while some have been removed, users should still be cautious. The other data issues noted here have been corrected.**

V2701001 and V2701002 something with the dates that R did not like.

**Corrected.**

V2701003 had no data from 2011 on (Table 1 states data from 2007-2013).

**Corrected in Table 1.**

V2701004, 50 cm VWC varies too much and no data present from 2011 on.

**Corrected in Table 1, and statement given about the possibility of fluctuations in the data.**

V2701005 data appears reasonable in values and range. No data present from 2010 on.

**Corrected in Table 1.**

V2701006 data appears reasonable in values and range. No data present from 2011 on.

**Corrected in Table 1.**

V2701023, 50 cm probe has high range of daily fluctuations and values greater than 0.60 m3/m3.

**Additional removals have been made.**

V2701025, 50 cm depth has a strange fluctuation that when plotted over the season it appears to have three distinct sets of data. This is not seen for the other sensors of this site. Many dielectric values are high and some sensors show very little response to seasonal rains or drying events.

**Further review of the site was completed and any further removals were completed.**

V2701034 has values > 0.60 at the 20 cm depth. At 50 cm depth moisture readings indicate saturation (0.50 or higher).

**Further review of the site was completed and additional removals have been made.**

V2701035 has values > 0.60 at the 50 cm depth.

**As stated in other comments, certain circumstances will result in soil moisture > 0.60. The ranges identified in Table 3 are only guidelines for the manual QAQC process.**

**Anonymous Referee #2**

Specific Comments: This article is well written and well organized. These data are clearly described and well formatted (Excel). The article targets an important issue of calibration and validation of growing space-based observations and hydrological model outputs against ground-based measurements. It is crucial to evaluate the reliability of those products before routine use at a global scale. However, there are a few points in the article that can benefit from improvement:

- 75: It would be clearer if the loam calibration equation were included.

**The loam equation is now included.**

- It is not clear why the authors have chosen only 11 years and whether this dataset will be continued or the operations has stopped after 2017

**Statement added in the Data Availability section for access to data beyond 2017.**

- If the stations are actively reporting the measurements, will the dataset be publicly available later? In addition, how long does it take data to become publicly available after ingest?

**Statement added in the Data Availability section for access to data beyond 2017. As stated in the Quality Control Process and Data section, automatic review can be completed in near real time, while secondary manual review is completed as needed or seasonally.**

- Since the network was designed for validation purposes, a comparison between the quality controlled data and existing datasets like SMAP and SMOS could be beneficial.

**Assessments of this type have been completed by other groups, such as Champagne et al. (2016) which assessed the SMOS and Aquarius products, and Chan et al. (2016) which gives initial results of the SMAP products. References to a selection of these works is now included in the introduction.**

- 94: Include the equation and the reference in this section

**Included and referenced.**

- 99-101: Then what is the range of the uncertainty involved in these calculations?

**Further details were provided in this section giving RMSE values found by a previous study and the continued issues in calibration method.**

-102: Providing these equations in the text will make it easier for the readers to follow your method.

**The general equation has been added and the sentence in question has been clarified.**

- 104-112: The issue is explained very well, but the authors do not clarify whether they have removed such problematic measurements from the data or if they are just recorded as they are.

**Sentence entered clarifying that these issues are removed from the dataset.**

- The sources of errors in the dataset are explained, but the study will benefit from a calculated estimate of such errors.

**Each station only has one probe at each location, so a quantitative estimation of errors cannot be completed.**

Technical Corrections:

1. 12: Change ESA to European Space Agency.

**Changed.**

2. 14-15: According to Fig. 1's scale, the two domains are 10x10 (100km2) and 40x40 (1600 km2)

**Corrected.**

3. 14-15: Please clarify the wording, because it is not clear if this is describing two different domains or two different domains with different spatial resolution among the sensors. The wording is not clear but the figure clearly shows an outer domain and a higher-resolution inner domain.

**The abstract has been reworded to clarify.**

4. 35-36: The last sentence need more explanation: "The high resolution of the network sites allows for both intergrid and intragrid validation".

**Rephrased.**

5. 59-61: The author clearly states that AAFC stations are located within the pasture sections but it is not clear what type of landscape the ECCC and University of Guelph cover. Please clarify this in the text.

**Details of the landscapes of the ECCC and UG sites have been noted in the Network Description section and Soil Moisture and Precipitation Site Details section.**

6. 63: Please refer to comment #2

**Corrected.**

7. 64: "45 x 55 km" should change to "45 x 55 km2 " and also x should be replaced my multiplication symbol

**This was corrected to 45 km by 55 km.**

8. 68: please refer to comment#2. Also there is a typo in km2-

**Both corrected.**

9. 74: "is" should be replaced by "are"

**Corrected.**

10. 79: "regularly" should be explained in detail. How often are the sites visited?

**Sentence added to explain the normal interval of visits.**

11. 80: "more frequently". Please refer to comment# 10

**Sentence added to explain the normal interval of visits.**

12. 99: adding a comma after "manufacturer supplied" will make the sentence more clear

**Rephrased.**

13. 126: "10 km2" should be replaced by "10 km"

**Corrected.**

14. 137: keep the consistency between the used words: year-round (line 40)

**Rephrased.**

15. 138: "occur" should be replaced by "occurring"

**Rephrased.**

16. 140: How do the thunderstorms producing solid precipitation (e.g. hail-stones) in the growing season will add to the error of your measurements?

**The sentence in question is in support of why winter and shoulder season data is being excluded from the dataset. Additional details on the impact of hail on the tipping bucket rain gauge data have been included in the Precipitation Instrumentation section.**

17. 47-50: what is the source of these thresholds? Please add a reference.

**References included.**

18. 154: What about irrigation. The abstract mentioned that the site is an agricultural site with croplands but irrigation is not mentioned anywhere in the text.

**Details about irrigation in the area added in the Network Description section.**

19. 182-186: The external funding sources for these operations should be mentioned in the acknowledgement.

**Funding sources for the University of Guelph have been added.**

20. 226-231: The references are in alphabetical and chronological orders. Rowlandson et al. 2013 should precede Rowlandson et al. 2015

**Corrected.**

21. 258-260: please refer to comment # 2.

**Corrected.**

**Anonymous Referee #3**

Specific comments

Introduction: the introduction is rather short and some elements could be added to better insist on the need of detailed hydrometeorological dataset in this region of Canada. For example, at P2 L 28-30, it would be interesting to add a few sentences and references on the remote sensing of soil moisture and the associated challenges, including the need for calibration at reference sites.

**The introduction was expanded to give further background on how the data has been used and many of those publications go into much greater depth on the challenges of both remote sensing of soil moisture and modelling.**

The second paragraph could be also more accurate:

- What is the "unique combination of landscape and climatic conditions" mentioned at P 2 L 33-34?

**The introduction was reworked and this phrase removed.**

- What are the other few existing monitoring networks available in the Canadian Prairies?

Over the last few years, these networks have been used to evaluate land surface models applied for hydrological and weather forecasting in Canada (Garnaud et al., 2016,2017).

**The other networks across the Canadian Prairies have been noted in the Introduction.**

P 2 L 40-41: the authors mention the presence of an eddy-covariance tower (also mentioned in the abstract). No additional information are available in the rest of the text. Are the data of this tower available from one of the institutional partner involved in this community site? If not, can the author add this dataset to their paper and make this dataset available on the FRDR website? It would be extremely valuable for the evaluation of land surface and hydrological models.

**Details have been added at end of the Network Description about the status of the eddy-covariance tower data.**

P 3 L 56: how does the University of Saskatchewan contribute to the community site?

**Detail added to end of section indicating U of S contributions.**

P 3 L 57-58: the AAFC stations are of potential interest for any studies in this area. I recommend the author to mention in the text the number of AAFC stations located in the area and to show their location on Figure 1.

**The figure has been modified to include the locations of the AAFC stations and the number of stations is now included in the Network Description section.**

P 3 L 63: the spatial scales are not accurate when compared to Fig. 1. Are the authors mentioning an area of 10*10 km2 instead of 10 km2? And 40*40 km2 instead of 40 km2? The abstract should be modified accordingly.

**Corrected.**

P 3 L 63-65: it would be interesting for the readers to know the number of stations in the network and to refer to Table 1 to mention that the exact position of each stations is given in Table 1.

**A specific number has not been given for the number of stations in the network as this total has varied over time. This has been clarified at the beginning of the section.**

P 3 L 75-78: which institutional partner has collected the additional data? Can these data be obtained on request?

**Various partners were involved in ancillary data collection and details of how to request this additional data are included in the paper.**

P 3 L 79-81: what is the typical frequency of the visits at the sites in summer time?

**Sentence added to explain the normal interval of visits.**

P 4 L 100: the loam calibration equation should be given in the paper since it is a paper focusing on the data.

**Equation has been included.**

P 4 L 101-102: what is the impact of using in-situ calibration equations on the accuracy of the computation of soil moisture? How large is the decrease in accuracy when using the loam calibration equation? Even if the in-situ calibration equations are not available for each probe, this comparison would be very useful for the reader to better understand the accuracy of the dataset presented in this paper.

**RMSE values from a calibration comparison project were provided in the text and further clarification was given regarding what calibration equation was used for the data of this publication.**

P 4 L 103-109: which treatment is applied to the soil moisture data when measurements issues occur with the Hydra Probe? Based on Table 3, it seems that these issues are identified during the automatic QC but Section 4.2 does not detail the final treatment applied to the soil moisture data.

**Additional clarification on how specific issues are dealt with in the dataset has been included**

P 5 L 125-126: what is the impact of the replacement of the rain gauges at the ECCC sites on the quality and the consistency of the times series of precipitation?

**The manufacturers listed accuracy for each type of TBRG has been listed to give a guideline on the uncertainty between gauges. Analysis of the aforementioned impact has not been completed.**

P 6 L 149-150: it is not clear in Section 4.2 how the flags described in Table 3 are used during the manual review process. I recommend the authors to describe the specific treatment applied on the dataset for each flag.

**Additional details describing the purpose of each flag has been included in Section 4.2, clarifying that the flags in Table 3 are not strict thresholds but guidelines to assist in QAQC.**

P 7 L 172: mention the format of the data on the FRDR website,

**Added statement in Data Availability section indicating that the data is available as a comma-separated-value format.**

P 7 L 179-180: is the data acquisition still continuing at the ECCC stations and at the University of Guelph stations? If it is the case, will the data be made available in a near future? At which frequency and on which platform? The fact that the data acquisition is still continuing is really important and should be clearly mentioned in the conclusion and also in the introduction.

**Multiple additions made to several sections clarifying that the future data can be requested from the corresponding author and that the network is still active.**

P 13: do the stations equipped with 4 probes share the same configuration? In particular, are the probes at 20 cm and 50 cm always located in the ground below the agricultural field as shown on Fig. 3?

**An additional figure has been included to clarify the location of the probes at each type of site and text has been added to the Soil Moisture and Precipitation Site Details section.**

P 14: Table 1: it would be interesting to know in the table which stations belong to ECCC and which stations belong to the University of Guelph.

**Table 1 has been updated to include these details.**

Technical Comments

Abstract

L 9: mention that Saskatchewan is located in Canada.

**Added.**

P 4 L 84: the references are not correct.

**Corrected.**

Comments on the dataset

Metadata

the metadata on the FRDR website does not contain the location of each stations. These information are only given in Table 1 of the submitted paper. I recommend the authors to add an ascii file containing the locations of each station. This file could have the same content as Table 1 and could be used with Python or R by a person interested in this dataset.

**A .csv version of Table 1 is now included with the metadata.**

[revised manuscript text omitted]

**Formatted Table**

**Formatted Table**